# Leading Edge: Intratumor Delivery of Monoclonal Antibodies for the Treatment of Solid Tumors

**DOI:** 10.3390/ijms24032676

**Published:** 2023-01-31

**Authors:** Ester Blanco, Luisa Chocarro, Leticia Fernández-Rubio, Ana Bocanegra, Hugo Arasanz, Miriam Echaide, Maider Garnica, Sergio Piñeiro-Hermida, Grazyna Kochan, David Escors

**Affiliations:** 1Oncoimmunology Unit, Navarrabiomed-Fundación Miguel Servet, Instituto de Investigación Sanitaria de Navarra (IdiSNA), Hospital Universitario de Navarra (HUN), Universidad Pública de Navarra (UPNA), 31008 Pamplona, Spain; 2Division of Gene Therapy and Regulation of Gene Expression, Instituto de Investigación Sanitaria de Navarra (IdISNA), Cima Universidad de Navarra, 31008 Pamplona, Spain; 3Medical Oncology Unit, Instituto de Investigación Sanitaria de Navarra (IdiSNA), Hospital Universitario de Navarra (HUN), 31008 Pamplona, Spain

**Keywords:** monoclonal antibodies (mAbs), intratumoral therapy, viral therapy, non-viral therapy

## Abstract

Immunotherapies based on immune checkpoint blockade have shown remarkable clinical outcomes and durable responses in patients with many tumor types. Nevertheless, these therapies lack efficacy in most cancer patients, even causing severe adverse events in a small subset of patients, such as inflammatory disorders and hyper-progressive disease. To diminish the risk of developing serious toxicities, intratumor delivery of monoclonal antibodies could be a solution. Encouraging results have been shown in both preclinical and clinical studies. Thus, intratumor immunotherapy as a new strategy may retain efficacy while increasing safety. This approach is still an exploratory frontier in cancer research and opens up new possibilities for next-generation personalized medicine. Local intratumor delivery can be achieved through many means, but an attractive approach is the use of gene therapy vectors expressing mAbs inside the tumor mass. Here, we summarize basic, translational, and clinical results of intratumor mAb delivery, together with descriptions of non-viral and viral strategies for mAb delivery in preclinical and clinical development. Currently, this is an expanding research subject that will surely play a key role in the future of oncology.

## 1. Therapeutic Antibodies: Beyond Conventional Monoclonal Antibodies (mAbs)

In 1988, Greg Winter and his team pioneered the technique to humanize monoclonal antibodies, and since then the field of personalized therapy using mAbs has been successfully developed for the treatment of various cancers [1]. To date, approximately more than 100 mAbs have been approved by the US Food and Drug Administration (FDA) for the treatment of cancer [2].

Monoclonal antibodies are engineered to specifically bind target antigens with high affinity. These antibodies then have the potential to induce complement-dependent cytotoxicity and promote innate and adaptive immune responses depending on their target specificities [3]. MAbs are large immunoglobulin molecules, usually in the range of about 150 kDa. The standard mAb molecule is composed of four polypeptide chains, forming a Y-shaped macromolecule complex. More specifically, mAbs are made of two identical heavy chains and two identical light chains connected by interchain disulphide bonds and non-covalent interactions. The light chain and two N-terminal domains of the heavy chain comprise the antigen-binding fragment (Fab). The variable domains of the heavy (VH) and light (VL) chains form the antigen recognition domain by adopting a structure with three hypervariable loops in each domain that comprise the complementary-determining regions (CDRs 1, 2, and 3), flanked by four conserved framework regions (FRs 1, 2, 3, and 4) (Figure 1A). The immune-mediated functions, such as antibody-dependent cellular cytotoxicity (ADCC) or complement-dependent cytotoxicity (CDC), are conferred by the C-terminal halves of the heavy chains, which comprise the crystallizable fragment (Fc). The conserved domains of the heavy chains differ between distinct antibody classes (isotypes). From these, the immunoglobulin G (IgG) isotypes are the main ones in human plasma. Hence, IgG-based monoclonal antibodies (mAbs) have become a dominant class of biotherapeutics in recent decades [4].

In the last decade, mAbs that block immune checkpoint molecules have shown remarkable clinical outcomes and durable responses in treated cancer patients. These mAbs have been engineered to reactivate antitumor immunity by blockade of T cell inhibitory molecules (immune checkpoints, IC), such as CTLA-4, PD-1, PD-L1, and LAG-3, which are the most studied IC. These molecules are expressed in different immune or non-immune cells. PD-L1 can be expressed by cancer cells and myeloid cells [5]. In addition, antigen-presenting cells (APC) express CTLA-4, and T cells and NK cells express PD-1, LAG-3, and other molecules, such as TIM-3 and TIGIT [6].

Antibodies targeting these latter targets are widely used for the management of cancer [7,8,9,10,11,12,13]. These therapies rely on systemic administration of these mAbs every three to four weeks. Moreover, these mAbs frequently present poor tissue penetration, intrinsic immunogenicity, and high production costs. All these issues suggest that there is still room for improvement. In fact, the large and complex structure of these mAbs causes several disadvantages: (1) high production cost due to their complexity and posttranslational modifications, (2) limited physicochemical stability, (3) systemic administration, which may cause off-target immune-related toxicities, (4) low penetration in solid tumors, and (5) poor penetration in the brain [14,15]. In certain cases, the production and use of full-length antibodies may be problematic, as mentioned above. Recent advances in antibody engineering have facilitated the production of a collection of antibody variants for the use in cancer, including other antibody formats (e.g., antibody fragments, bispecific antibodies (BsAbs), and non-IgG scaffold proteins) and antibody derivatives (e.g., antibody–drug conjugates (ADCs) and immunocytokines) [16]. Right now, the field of antibodies is revolving around the engineering of the antigen-binding region from heavy single-chain antibodies (HCAbs), such as those from camelids (dromedaries, camels, llamas, alpacas, guanacos, and vicuñas), coined with the term “nanobodies” [17]. Compared to conventional mAbs, HCAbs are homodimers made of two identical heavy-chain molecules, lacking both the light chain and the constant domain 1 (CH1) of the heavy chain. These antibodies possess a significantly smaller molecular mass, of approximately 95 kDa (Figure 1B). The variable antigen-binding domain of HCAbs (called VHH) retains full antigen-binding potential despite lacking the light chain. Indeed, the VHH domain is the smallest naturally occurring antigen-binding fragment [18].

The discovery of HCAbs has spurred the development of nanobody production platforms due to their inherent properties, which make them very attractive tools for cancer treatment: (1) nanoscale dimensions enable deeper tumor penetration, (2) certain nanobodies can cross the blood–brain barrier (BBB) [19], and (3) high affinity and specificity for their targets with low off-target accumulation. Structurally, nanobodies are similar to the VH domain of conventional antibodies, with four FRs and three CDRs. Besides, nanobodies exhibit high homology with the VH domain of human family III immunoglobulins. In addition, nanobodies present higher hydrophilicity, with increased solubility and enhanced physicochemical stability compared to the variable fragments from conventional antibodies. Moreover, the CDR3 motif tends to be longer, being the main region implicated in antigen binding. The antigen-binding region is made by CDR3 and CDR2, together with some FR residues. This extended CDR3 loop allows binding to small cavities or concave epitopes (mainly conformational epitopes), such as catalytic sites of enzymes. In contrast, conventional antibodies are better at recognizing small chemical groups (haptens), peptides, or flat epitopes on proteins [20,21]. Currently, several nanobody-based therapeutics are under clinical trials for the treatment of a variety of diseases, including cancer, autoimmune diseases, and viral infections [22]. For solid tumors, PD-L1 and CTLA-4 are the most common targets, which are summarized in Table 1.

## 2. In Vivo mAb Gene Delivery Systems

Nowadays, more than 168 interventional clinical trials are evaluating the delivery of mAbs in solid tumors. The majority are against PD-1 or bispecific for two targets, which are summarized in Table 2. New engineering platforms are arising to improve therapeutic mAbs. These platforms implement strategies to optimize the potency, efficiency, and stability of mAbs, as well as improving cell manufacturing, large-scale development, and delivery for clinical application. The intrinsic biochemical properties of antibody sequences are frequently incompatible for large-scale manufacturing. This fact strongly limits the systematic implementation of mAbs into clinical practice [23]. Moreover, intravenous in vivo administration of mAbs must be carried out at high doses to achieve therapeutic efficacy, which sharply increases the cost of therapies [24]. For these reasons, novel approaches for in vivo delivery of mAb-based products need to be implemented.

The first approach is to optimize mAb complementary DNA (cDNA) sequences for expression from specific vectors. One of such approaches relies on optimization for expression from adeno-associated virus (AAV). Nevertheless, since the use of mRNA vaccines to tackle COVID-19 [25,26], non-viral synthetic nucleic acids are emerging as promising expression vectors of any encoded sequence. These approaches can be based on synthetic DNA formulated for facilitated delivery by electroporation, or lipid nanoparticle (LNP)-encapsulated messenger RNA (mRNA) [27,28,29]. Since no purification of the final product is needed, the mAbs are directly expressed within the in vivo-targeted cells, leading to antibody secretion into the systemic circulation. In the case of RNA expression systems, which can include the use of RNA replicons such as those from Semliki Forest virus (SFV), the mRNA is directly translated within the cell, avoiding the nuclear steps used by AAV and DNA delivery systems (Figure 2).

Plasmid DNA-encoded mAbs (pDNA-mAbs) can also be engineered as carriers of synthetic antibody genes. Several studies demonstrated that pDNA-mAbs expression leads to consistent serum concentrations for up to 2–3 months, reaching a maximum 2 weeks following administration. However, an important drawback is the relatively quick loss of the vector [30,31]. Another important issue is that synthetic DNA requires efficient delivery systems. Portable electroporation systems may represent a good option for pDNA-mAbs delivery, as shown in preclinical models. For clinical use, CELLECTRA™, Ichor TriGrid, and Igea Cliniporator (IGEA) systems are currently used [30,32].

Recently, and following the application of mRNA vaccines, mRNA platforms have shown very good clinical results as fast and efficient delivery systems. Nevertheless, for long-term delivery, repeated administrations are required due to its short half-life [33]. Two RNA types have been proposed for application in cancer immunotherapies: The first one consists of the conventional mRNA containing an open reading frame (ORF) flanked by 5′ and 3′ untranslated regions (UTRs). The second type corresponds to self-amplifying RNAs (saRNA) derived from the positive-stranded alphavirus RNA genome, also called replicons [34]. For mRNA delivery, the vector formulation is key due to the intrinsic instability of RNA. The current lipid nanoparticle formulations (LNP) improve RNA stability and delivery into target cells. An example of these formulations are the mRNA vaccines applied for COVID-19 [35]. Replicons are delivered through their natural viral vectors. These self-replicating RNAs contain all viral information for the replicase but lack virus genes for assembly and propagation. An example of these are alphavirus-based vectors, such as Semliki Forest virus (SFV) and coronavirus-based replicons [36], among others.

Due to the success of mRNA vaccines in widespread human immunization schemes, the use of RNA-based delivery systems has gained much attention.

## 3. Routes of mAbs Administration

The success of immune checkpoint blockade immunotherapies has demonstrated significant advances in the treatment of many cancers. However, current immunotherapies fail in most cancer patients. There are many reasons for the failure of immunotherapies, which include poor immunogenicity characterized by reduced tumor infiltration with immune cells, and systemic immune dysfunctionality [9,37,38,39]. The lack of penetration of mAbs within the tumor environment adds up to the poor immunogenicity of many cancer types. These problems contribute to the failure of ICB therapies in many cancer patients.

The systemic parenteral use of therapeutic mAbs has unequivocal advantages, allowing simplicity of administration and predictable serum pharmacokinetics. However, this mode of delivery presents limitations and disadvantages, which include poor penetration into solid tumors and systemic toxicities caused by off-target effects, with systemic inflammation and autoimmune or autoimmune-like reactions [40,41]. Indeed, serious immune-related adverse events (irAEs) are associated with recurrent systemic administration of ICB antibodies [42,43,44]. These constraints could be overcome, or at least reduced, by enhancing mAbs availability within the tumor microenvironment (TME). One way to achieve this would be through intratumor delivery of mAbs and locoregional delivery.

Several routes have been tested for mAbs delivery, for example subcutaneous (SC), intramuscular (IM), oral, and intratumor (I.T) administration. SC injection would be the most convenient for patients. Thus, various mAbs have emerged, designed for subcutaneous administration [45,46]. This administration route would be suited for self-administration by the patient, although so far, the accuracy and efficacy of mAbs delivered in such a way is hard to predict. Indeed, this is a major issue for human therapy. In addition, the mAb formulation itself for SC delivery is still a major challenge for drug development [47]. The same arguments can be applied for muscular delivery. Nevertheless, the clinical application of these type of drugs for these administration routes is hampered by their complex structure. Oral administration of mAbs has been discarded because of inefficient transport through tissue barriers such as the intestinal mucosa, while they are quickly degraded by proteases in the gastrointestinal tract. In contrast to other strategies, intratumor delivery can increase the therapeutic index of mAbs by restricting them to within the tumor environment, with a reduced risk for off-target toxicities. In addition, repeated intratumor injections can be administered, leading to much higher local bioactive drug concentrations [48,49].

Despite the challenges associated with intratumor delivery, it has substantial potential to improve immunotherapies.

## 4. Intratumor mAbs Delivery in Solid Tumors

Intratumor administration using image-guided injection is achievable for most organs [48]. Following administration, the therapeutic agents first diffuse throughout the injected area, thereby achieving a very high local concentration. Overtime, drugs will dissipate into systemic circulation. In fact, this gradual absorption into the blood can have pharmacokinetic advantages that will permit higher local doses with better tolerability, as shown for other protein-based drugs [50] (NCT02304393). Importantly, intratumor delivery allows the immediate access to tumor-draining lymph nodes and other lymphoid structures within the tumor tissue, amplifying the immune response [51,52,53]. Importantly, intratumor administration of mAbs and other drugs shows an abscopal effect in distal metastases, indicating that this administration route can have systemic activities [54,55]. Injections in multiple tumor lesions within the same patient can also enhance polyclonal responses, despite the high variability of cancer cells in metastases [56]. In addition, several studies have evaluated intratumoral mAbs and short peptides against receptors expressed on the blood–brain barrier (BBB) delivery to achieve brain targeting [57,58].

Several drug combinations have been evaluated with success in preclinical studies, for example combinations of antibodies targeting CTLA4, OX40, PD-1, and CD137 [59]. Some formulations also allow a slow intratumor release of mAbs, which leads to a prolonged and improved therapeutic index [60].

There are two main vector strategies for optimized intratumor mAbs administration: non-viral- and viral-based vectors. Both strategies present intrinsic advantages and disadvantages (Figure 3). The efficiency of modifying host cells with mAb-encoding nucleic acids is better with viral vectors, but a major drawback of viral vectors is their immunogenicity and potential cytotoxicity [61,62]. Replicative oncolytic viruses are also emerging as promising anti-cancer treatments, but in this case, these can be regarded as an in-situ treatment that releases antigens and damage-associated molecules, rather than virus-based vectors for drug delivery. On the other hand, non-viral vectors possess better safety profiles but less efficient capacities to modify target cells in vivo [63,64].

Several preclinical and clinical trials are testing strategies to deliver mAbs locally within the tumor. This review focuses on the current preclinical and clinical achievements in mAbs intratumoral delivery using non-viral and viral vectors for the treatment of solid tumors.

## 5. Preclinical Non-Viral Vectors for mAbs Intratumor Delivery

There are a variety of delivery vehicles and scaffolds that have been engineered over time to transport biomolecules, including mAbs [65,66]. All are designed based on enhancing biodistribution within the tumors, and some have been adapted to operate within the physicochemical properties of the TME, such as a low pH or high concentrations of ATP, while sustaining their cargo release.

### 5.1. Nanoparticles and Lipid Vesicles

The most widely used carriers include polymer nanoparticles (NPs), inorganic NPs, and lipid-based NPs for drug delivery. The main goal for utilizing NPs is to improve the bioavailability of immunotherapeutic agents while reducing toxicity. Typically, lipid nanoparticle formulations are composed of pH-responsive lipids or cationic lipids bearing tertiary or quaternary amines to encapsulate the polyanionic RNA molecules. In addition to this main composition, nanoparticles incorporate other neutral helper lipids to constitute a hydrophilic layer over the nanoparticles to stabilize the lipid bilayer and enhance RNA delivery [67].

MnCaCO_3_/ICG nanoparticles have been produced loaded with PD-L1-targeted siRNA that can be intravenously injected. PD-L1 is one of the main T-cell inhibitory molecules expressed by cancer cells, through biding to PD-1 expressed on the T cell surface [68,69]. In addition, PD-L1 expression also confers cancer cells with resistance to apoptosis. Hence, PD-L1 silencing combined with photodynamic therapy (PDT) showed powerful antitumor effects [70]. These nanoparticles for the treatment of cancer have been used to treat recurrence after surgical resection. A fibrin gel was used to encapsulate calcium carbonate nanoparticles pre-loaded with anti-CD47 antibody and applied locally in the tumor. This treatment achieved polarization of tumor-associated macrophages towards M1-like phenotypes, leading to tumor control both locally and distally after surgery [71]. Another interesting strategy is based on the use of PD-1-positive tumor-derived vesicles to disrupt PD-1/PD-L1 interactions [72]. Recently, it has been shown that PD-L1 present on tumor cell-derived extracellular vesicles (sEVs) play a key role in immunosuppression and resistance to immunotherapies. Therefore, counteracting these vesicles could also improve conventional treatments [73].

Moreover, polyethylenimine (PEI) is a cationic polymer that has been extensively used for intratumoral delivery. PEI can be modified with cholesterol or other lipoic acids to improve gene delivery. In fact, a potent nanoplexed formulation with Poly I:C complexed with PEI was recently developed. The powerful antitumoral activity in murine models led to clinical evaluation [74] (NCT02828098).

### 5.2. Microneedle Delivery Platforms

Microneedles (MNs) have become a leading delivery strategy for transdermal drug administration and have been reconverted for immunotherapies. MNs are micron-sized and minimally invasive. These microneedles facilitate transdermal local delivery of different cargoes, from proteins to small molecules. This procedure achieves controlled and sustained cargo release [75]. Microneedle (MN) patches can also be formulated to modulate the TME, for example biodegradable and pH-sensitive MNs, or the MN-based GOx/CAT enzymatic system. Wang and colleagues developed a new procedure to perform localized delivery of anti-PD-1 for melanoma treatment, in which MN was integrated with pH-sensitive dextran nanoparticles loaded with glucose oxidase (GOx). This mechanism of delivery is PH-dependent. A decrease in pH promotes self-degradation of the nanoparticles within the MN, allowing a continuous release of mAb within the tumor environment. This intratumoral strategy showed efficacious tumor growth inhibition in vivo in a mouse melanoma model [76].

In addition, MN tools allow intratumoral co-delivery of two or more ICB agents to achieve synergistic therapeutic effects. MNs based on the GOx/CAT enzymatic system facilitated sustained release of ICB therapeutics, for example to block PD-1 and deliver an indoleamine 2,3-dioxygenase (IDO) inhibitor in a B16F10 mouse melanoma tumor model. A synergistic anti-tumor activity of IDO inhibition and PD-1 blockade was observed with prolonged survival [77]. Modifications of MN strategies can be performed with other approaches, for example implementation of MNs implemented with cold atmospheric plasma (CAP) to facilitate transdermal penetration of CAP to tumor tissues to induce immunogenic death. In this way, enhanced release of tumor-associated antigens was achieved to elicit dendritic cell maturation (DC) and T cell responses [78].

### 5.3. Hydrogels as Delivery Vehicles

Hydrogels are biomaterials formed by a cross-linked porous network of polymers. Some of these are termed ‘smart biomaterials’ when they have the property of changing their structural properties to respond to environmental stimuli (e.g., light, temperature, pressure, electric and/or magnetic fields, pH, solvent composition, and recognition of ions and specific molecules) [79,80,81]. Hydrogels have been proven to be non-toxic and biodegradable, becoming a potential vehicle for encapsulating therapeutic molecules. It has recently been shown in preclinical murine melanoma and breast cancer models that a PEG-b-poly(L-alanine) hydrogel permitted encapsulation and release of tumor lysate cells with granulocyte–macrophage colony stimulating factor (GMCSF), anti-PD-1, and anti-CTLA-4 simultaneously in the tumor. This co-delivery of a tumor vaccine and dual immune checkpoint inhibitors showed a significant increase of efficacy [82]. Combinatorial local immunotherapy with celecoxib and anti-PD-1 from a hydrogel system synergistically enhanced activated T cells, and reduced regulatory T cells (Tregs) and myeloid-derived suppressor cells (MDSCs) within the tumor microenvironment [83].

### 5.4. HSC−Platelet−Anti-PD-1 Assembly

Platelet engineering has surfaced as an interesting novel approach due to their unique targeting ability toward inflammation sites. Natural platelets have been shown to be conjugated with anti-PD-1 antibody for targeted delivery following tumor resection to inhibit tumor recurrence. A cell–cell combinatorial delivery platform was constructed based on conjugates of platelets and hematopoietic stem cells (HSCs) for leukemia treatment. With the homing ability of HSCs to the bone marrow, the HSC–platelet–anti-PD-1 assembly could effectively deliver the anti-PD-1 antibody in an acute myeloid leukemia mouse model [84].

### 5.5. Intratumor Plasmid DNA (pDNA) Electroporation

An emerging strategy to be applied in human therapy is electroporation of plasmid DNA. pDNA-based delivery is cost-efficient, allows for combination therapies, and presents low immune-related toxicity risks by intratumor gene electrotransfer. This strategy has been used for the combined delivery of plasmids encoding IL-12 and an anti-PD-1 antibody that induced good anti-tumor responses [85]. Other studies combined anti-CTLA-4 and anti-PD-1 antibodies to evaluate their pharmacokinetics and pharmacodynamics when delivered via intramuscular or intratumor electroporation in mice [86].

### 5.6. Antigen Peptides Conjugated on mAbs

New strategies based on engineered mAbs are appearing. Indeed, antigen/α-PD-L1 conjugate therapy showed a strong local antitumor immune response [87]. In line with similar works, an anti-PD-L1 peptide-conjugated prodrug nanoparticle (PD-NP) has been developed to avoid severe toxicity and improve antitumor activity of T cells in cancer immunotherapy [88].

## 6. Intratumor Delivery of mAbs with Viral Vectors in Preclinical Models

Delivery of therapeutic antibodies using gene therapy vectors has been in continuous progress for more than two decades [89]. From these, viral vectors are the most used delivery vehicles due to their remarkable gene delivery efficiency [61,90,91].

### 6.1. Oncolytic Viruses for the Treatment of Solid Tumors

Oncolytic virotherapy is arising as a promising strategy for several solid tumors. Its mechanisms of actions integrate specific infection and destruction of tumor cells and the modulation of the TME. Cancer cell death causes the release of danger signals, which initiate innate and adaptive immune responses [92]. Oncolytic viruses can be genetically modified to express proteins, decrease pathogenicity, increase lytic potential, and enhance immunogenicity, improving the risk–benefit ratio for clinical development [93]. A very relevant candidate for oncolytic virotherapy is the Newcastle disease virus (NDV). NDV is an enveloped, negative-sense, single-stranded RNA virus of the Paramyxoviridae family [86]. NDV induces activation of innate and adaptive antitumor responses in addition to prompting immunogenic cell death [94]. In a recent study, two NDVs expressing anti-PD-1 and anti-PDL1 were evaluated in a murine melanoma model following intratumor injection of these recombinant NDVs. Both elicited systemic antitumor responses, especially when combined with systemic CTLA4 checkpoint inhibition [95]. Another study demonstrated efficacy after intratumor administration of an engineered vaccinia virus encoding a single-chain variable fragment against TIGIT, together with systemic PD-1 or LAG-3 blockade [96]. One advantage of using oncolytic viruses combined with immune checkpoint blockade is the synergistic immunogenic action from viral replication [97]. Oncolytic virotherapy can upregulate PD-L1 expression in the TME through virus-induced type I and type II IFNs. This characteristic can be taken advantage of by expressing PD-L1- or PD-1-blocking antibodies. For example, herpes simplex virus (HSV) expressing a single-chain variable fragment (scFv) against PD-1 (aMPD-1 scFv) modulated the TME by releasing damage-associated molecular patterns. This promoted antigen cross-presentation, and infiltration by activated T cells [97]. Another example is the application of oncolytic adenovirus Ad5/3-Δ24a expressing the complete human mAb for CTLA-4 in xenograft mouse models. The local expression of the anti-CTLA4 antibody resulted in significantly higher concentrations within the tumor, while plasma levels remained at nontoxic concentrations [98].

### 6.2. Semliki Forest Virus (SFV)

Semliki Forest virus (SFV) is an alphavirus that contains a positive-strand RNA genome, which can be easily engineered to express transgenes of interest. SFV vectors have shown potent antitumor properties in a wide range of preclinical studies. These vectors overexpress heterologous genes at very high levels in a broad variety of cells, induce apoptosis in tumor cells, and stimulate IFN-I responses [99]. SFV vectors can be used directly as carriers of packaged RNA, or by introducing RNA or DNA into the cells, encoding their genetic information. SFV vectors express transgenes only for a short period of time because of their cytopathic nature. This can be used as an advantage, and as such, SFV demonstrates satisfactory results when compared to other viral vectors in preclinical studies [100,101,102]. SFV has been successfully used for local IL-12 expression, a treatment with high efficacy for antitumor immunity in mouse models [103,104]. SFV vectors have been used for local transient expression of immunomodulatory mAbs. A short local expression of anti-PD-L1 mAbs from a SFV vector showed significant antitumor efficacy in a colon adenocarcinoma model [103]. Moreover, several approaches have been developed for local or intratumor genomic RNA delivery to utilize intracellular replicons without the need for purifying virus-like particles [105,106].

### 6.3. Adeno-Associated Viral Vectors

Adeno-associated viral vectors have strongly emerged as key therapeutic tools for in vivo gene delivery in human gene therapies for the treatment of many disorders [107,108]. AAVs are non-enveloped, non-pathogenic viruses belonging to the Parvoviridae family, which require coinfection by a helper virus, such as adenoviruses, for their replication and propagation [107]. The use of AAV-based vectors represents an attractive strategy for their excellent safety profile, high transgene stability, and feasible production. Furthermore, they have shown high transduction efficiency in various target tissues, transducing both quiescent and dividing cells. This property allows long-term transgene expression. A major drawback is their limited packaging capacity, of about 5 kb. In addition, these viruses are very immunogenic, and generate strong immune responses that limit their use in therapy [109,110]. To counteract this issue, AAV capsid engineering strategies have been introduced for repeated use of AAVs. AAV applications are being extended for targeted delivery of ICIs with tumor-targeted AAV vectors. Hence, a PD-1-specific scFv-Fc fusion protein was delivered by a Her2-targeted AAV vector, leading to reduced tumor growth in mouse models [111]. AAV vectors provide sustained nanobody expression both locally and systemically in preclinical models of human diseases, including solid tumors [112,113].

### 6.4. Adenovirus Vectors

Adenoviruses have been extensively used in gene therapy for many decades. Adenoviruses constitute a large family isolated from a broad range of hosts. The adenovirus virion contains a double-stranded DNA genome of about 26–45 kilobases, which causes self-limiting mild infections, usually without clinical symptoms. Adenoviruses can efficiently infect resting and dividing cells, and their genome remains as an episome within the nucleus, without integration. Adenoviruses are also oncolytic for cancer cells. They have been used to express an anti-PD-L1 antibody from a mifepristone-inducible expression system. Local administration of this high-capacity adenoviral vector (HCA-EFZP-aPDL1) in subcutaneous lesions led to a significant reduction in tumor growth, with minimal antibody release to the circulation [114].

## 7. Intratumor mAbs Delivery under Clinical Development

At present, more than 27 interventional clinical trials are evaluating intratumor delivery of mAbs (Table 3). Most of these are phase I/II trials and include heterogeneous cohorts of patients diagnosed with different tumor types with local lesions. Most ongoing clinical trials are based on delivery of a CD40 agonistic monoclonal antibody. This is an agonistic antibody that facilitates the recruitment of immune cells with anti-tumor capacities. This approach has been demonstrated to be effective in various preclinical models. However, treatments with human CD40 mAbs presented modest antitumor activity in cancer patients, characterized by low efficacy and dose-limiting toxicity when systemically administered. Nevertheless, preclinical studies have demonstrated that intratumor administration of CD40 agonists improves efficacy and reduces irAEs; mainly, irAEs associated with cytokine release syndrome and liver toxicity involving myeloid cells and platelets [115]. Intratumor administration of the anti-human CD40 mAb (ADC-1013) into superficial lesions showed, as a limitation, the impossibility of intratumor administration into vascularized tumors in internal organs such as the liver, spleen, and kidney [116]. Subsequent clinical trials implementing intratumor anti-CD40 therapy are ongoing (Table 3).

## 8. Selected Clinical Trials According to Therapy Types

### 8.1. Clinical Trials with Administration of Conventional Monoclonal Antibodies

CTLA-4 and PD-1 are the most widely used monoclonal antibodies for cancer treatment. As such, they have been preferably selected for intratumor administration (Table 1). Several ongoing clinical trials have provided preliminary evidence that injection of a combination of anti-CTLA4 and anti-PD-1 mAbs directly into sites of glioblastoma resection is safe. An example of these clinical trials is represented by the GlitIpNi study (NCT03233152). This is an interventional phase I, first-in-human, open-label study of intratumor administration of ipilimumab with systemic nivolumab in glioblastoma. The aim of this clinical trial is to exploit the potential synergy of combined intratumor anti-CTLA-4 mAb (ipilimumab) with systemic PD-1 blockade with nivolumab, while minimizing risks for immune-related toxicity of ipilimumab following resection of recurrent glioblastoma. Ipilimumab is administered at the end of the neurosurgical resection as a single dose. This methodology has been previously applied within the context of phase III clinical trials with sitimagene ceradenovec. Nivolumab will be administered intravenously 24 h prior to neurosurgical resection, with 5 additional doses on days 15, 29, 43, 57, and 71. The inclusion criteria include male or female patients aged ≥ 18 years with histopathological diagnosis of glioblastoma (WHO grade IV glioma of the central nervous system). A measurable tumor lesion is required, characterized by gadolinium enhancement on T1-MRI of the brain, with no evidence of clinically relevant spontaneous intra-tumor hemorrhage on baseline MRI or in prior history, with an ECOG performance status score of 0, 1, or 2. Other criteria include normal total serum bilirubin, AST, ALT, serum creatinine, and FT4 hormone concentrations, as well as normal absolute neutrophil counts, platelets, and hemoglobin concentration without growth factor support. The estimated study completion date is 17 November 2023.

### 8.2. Clinical Trials with Oncolytic Viruses

Several clinical trials are currently evaluating the safety and efficacy of intratumor anti-CTLA4 mAbs administration in combination with virotherapies, as exemplified by the ISI-JX study (NCT02977156).

This study is an interventional phase I, dose escalation, multicenter and open-label clinical trial, evaluating local anti-CTLA-4 blockade with ipilimumab in metastatic or advanced solid tumors. This local therapy is combined with pexastimogene devacirepvec (Pexa-Vec), an oncolytic virus genetically modified to express GM-CSF. This clinical study is a proof of concept consisting in two parts. In the dose selection part (part A), patients were treated with an intratumor injection with Pexa-Vec alone at week 1, followed by intratumor injections of Pexa-Vec plus ipilimumab (up to 4 dose levels) at weeks 3, 5, and 9. The second part of the clinical trial consists of an expansion of cohorts (part B) (up to 3 cohorts), in which patients were treated with an intratumor injection with Pexa-Vec alone at week 1, followed by intratumor injections of Pexa-Vec plus ipilimumab at weeks 3, 5, and 9. The inclusion criteria include male and female patients aged ≥ 18 at the time of inform consent signature, with histologically confirmed advanced/metastatic solid tumor refractory or relapsing from standard therapy. The study can also include patients that refused or did not tolerate the standard therapy. Any tumor types can be included for part A, except for hepatocellular carcinoma (HCC). In part B, tumor types may include melanoma, MSI-high colorectal carcinoma (CRC), head and neck tumors, gastric cancers, triple-negative breast cancers, and mesothelioma. Patients must have at least one injectable site ≥ 2 cm and ≤8 cm in diameter and one distant non-injected measurable site (target site). Intratumor injections were performed by a radiologist using imaging-guidance, ultrasound, or computed tomography (CT). The dose to be injected can be divided among 1 to 5 tumor lesions. The study is completed and awaiting the final analyses and conclusions.

Other clinical studies are assessing the safety and efficacy of oncolytic vectors expressing IL-12 combined with anti-PD-1 antibody, as exemplified by the MVR-C5252 study (NCT0509544). This is an interventional, first-in-human, phase I, open-label study of the recombinant oncolytic HSV-1 (C5252) expressing IL-12 combined with anti-PD-1 antibody therapy in patients with recurrent or progressive glioblastoma. C5252 is a genetically engineered oncolytic HSV-1 developed by ImmVira’s OvPENS, an FDA-regulated drug product. The estimated study start date will be March 2023. This is a first-in-human study of C5252 monotherapy designed to evaluate the safety and tolerability of a single intratumor injection of C5252 in patients with recurrent or progressive glioblastoma. Part 1 of the study is a dose escalation study for C5252. Approximately 36 evaluable participants will be enrolled, and further recruitment into the trial will depend on the observed toxicities and activity. Once the recommended dose (RD) is identified, a dose expansion part 2 study will be carried out with an increase in patients for the evaluation of safety, tolerability, and preliminary efficacy of a single intratumor injection of C5252. Inclusion criteria include male and female patients aged ≥ 18 with life expectancy > 12 weeks. Participants must have confirmed recurrent supratentorial glioblastoma following progression after at least 1 line but no more than 2 lines of therapy. Evidence of progression will be measured by RANO criteria based on MRI scans, and residual lesions must be ≥1.0 cm and <5.5 cm in diameter. In addition, participants must have normal organ and bone marrow function, and must commit to the use of a reliable method of birth control. The study status is not yet recruiting.

### 8.3. Clinical Trials with Non-Viral Lipid Nanoparticles

Most clinical trials with lipid nanoparticles are designed to deliver cytokine genes locally into lesions, but new clinical trials are evaluating nanoparticles to deliver mAbs such as pembrolizumab. For example, in the high-risk ductal carcinoma in situ (DCIS) study NCT02872025, which is an interventional early phase 1 clinical trial. This pilot study aims to investigate the change within the immune microenvironment within tumors in in situ high-risk ductal carcinoma (DCIS) after a short-term exposure to pembrolizumab. This study includes a dose escalation design, followed by a dose expansion phase at the maximum tolerated dose. The primary objective of the first phase is to assess the safety and feasibility of an intra-lesion pembrolizumab injection. The maximum tolerated dose is then used for the expansion phase. The expansion cohort contained a control group or the treatment group. The control group includes patients with a single surgery within a four-month timeframe following the diagnosis. The treatment group will consist of patients after four doses of intralesional pembrolizumab monotherapy prior to surgery (first five patients enrolled). The rest of the patients received 2 doses of intra-lesion pembrolizumab and intra-lesion mRNA-2752 3 weeks prior to surgery. The synthetic mRNA encodes OX40L, IL-23, and IL-36γ (mRNA-2416). All subjects in the expansion cohort were diagnosed by a baseline MRI and prior to surgery. Baseline and pre-surgical MRI images will be compared for changes in tumor volume. The escalation phase finished on 14 August 2018, and the study is currently in the expansion phase.

## 9. Key Notes and Conclusions

The use of conventional mAbs is limited by their complex structure. Novel designs such as bispecific antibodies or nanobodies can be good alternatives.Direct in vivo delivery of synthetic nucleic acids encoding antibodies such as plasmid DNA or mRNA platforms represent new approaches for in vivo delivery of antibody-like biologics. These platforms have advantages such as rapid product development and simpler manufacturing processes.Intratumor delivery increases efficacy, local bioavailability, reduces toxicity, and improves the antitumor immune responses.Intratumor delivery can be combined with other systemic strategies and can be implemented with non-viral- and viral-based delivery methods, including nanoparticles or lipid vesicles.Virotherapies are promising approaches for cancer treatment and as delivery vehicles of mAbs, but may pose biosafety concerns, especially for the use of oncolytic viruses.

## 10. Future Perspectives

Novel intratumoral immunotherapy strategies are steadily emerging for solid tumor treatment and will probably change the procedure of drug development in oncology. Nevertheless, these intratumoral strategies drive transformation in clinical practice and advance new possibilities for next-generation personalized medicine. In this review, we have shown the main strategies for intratumor mAbs delivery using non-viral and viral gene vectors expressing mAbs inside the tumor mass. Nowadays, there are more clinical trials with conventional mAbs, but with engineered DNA or mRNA platforms, the bispecific antibodies or nanobodies are being implemented in clinical trials. Indeed, with the approval of two mRNA LNP vaccines to prevent COVID-19, current preclinical and clinical trials are focused on synthetic mRNA. mRNA properties stand out for clinical applications, as they can encode multiple antigens, are non-integrative, and have rapid and scalable manufacturing. However, applying the best vehicle for mRNA administration was elaborated. Non-viral methods are safer, but mRNA biodistribution and potency is needed. In fact, several delivery vehicles and scaffolds have been engineered to enhance mAbs biodistribution within the tumors and increase their efficacy. Alternately, self-amplifying RNA (saRNA) derived from alphavirus expression vectors could be a promising approach. It has shown to be very efficient to induce humoral and cellular responses against many antigens in preclinical models, being superior to non-replicating mRNA and DNA.

Currently, nanobodies are emerging as a new generation of antibodies. Indeed, nanobodies bind their antigens quickly and specifically, resulting in high action soon after their administration. Thus, the application of nanobody dimers or multimers could be applied to enhance antitumor activity in solid tumors. Nevertheless, nanobodies targeting immune checkpoints are mainly concentrated in PD-1/PD-L1. Hence, it is necessary to expand more research on targeting other immune checkpoints.

## Figures and Tables

**Figure 1 ijms-24-02676-f001:**
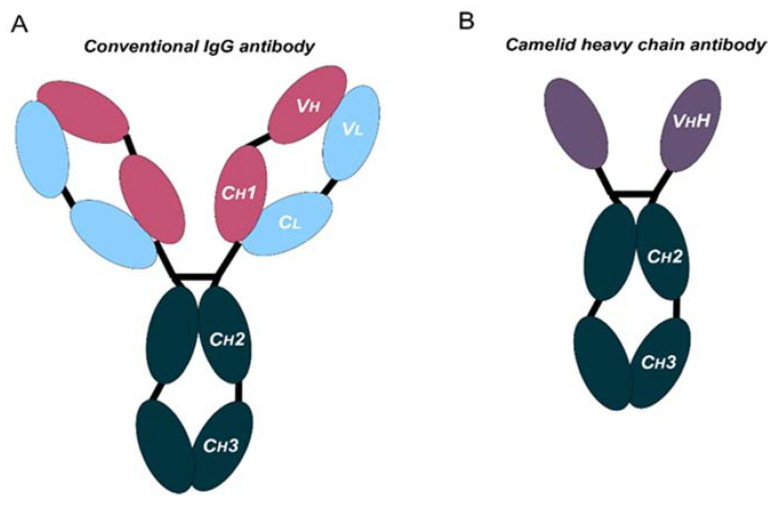
Schematic representation of the structure of conventional and camelid heavy-chain antibodies. (**A**) Structure of a conventional IgG antibody, composed by two heavy and two light chains. Conserved and variable heavy-chain domains (Ch, Vh) and light domains (Cl, Vl) are indicated. (**B**) Structure of a camelid heavy-chain antibody, composed by two heavy chains.

**Figure 2 ijms-24-02676-f002:**
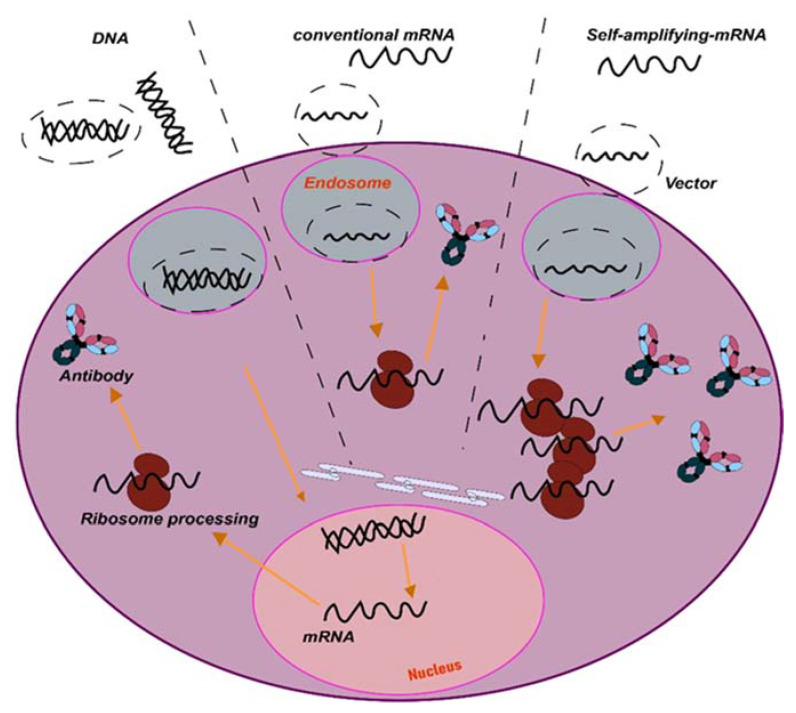
Schematic representation of nucleic acid delivery systems. Liposome-transported DNA is internalized through endocytosis, or by physical means such as electroporation. Inside the cell, DNA is imported into the nucleus, where it will be transcribed. Both conventional and self-amplifying RNAs require a delivery system for cell uptake by endocytosis, which it is followed by release from the endosome into the cytoplasm. Then, the mRNA is immediately translated. For replicons, the RNA is translated into replicase proteins that will auto-amplify the RNA, including antigen-encoding subgenomic mRNAs, leading to large quantities of the encoded antigen.

**Figure 3 ijms-24-02676-f003:**
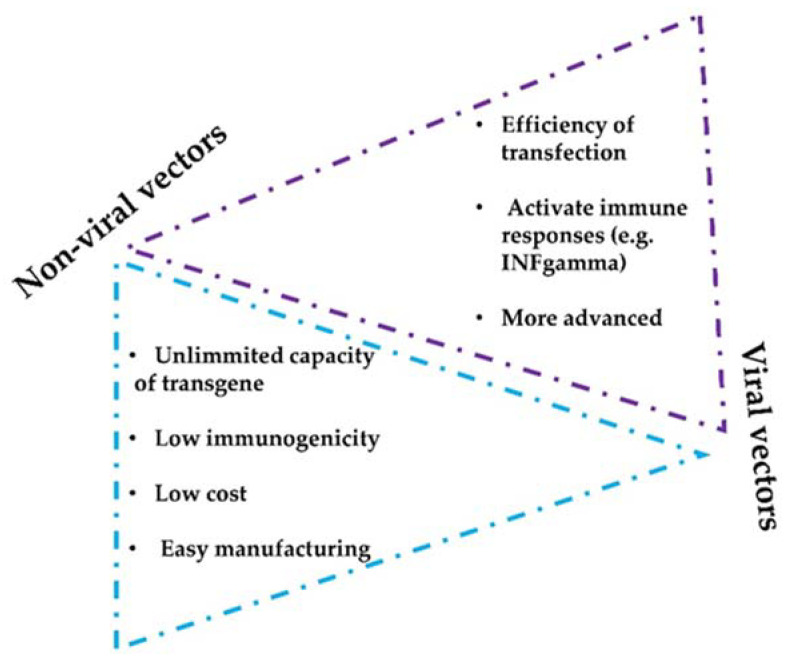
Considerations of non-viral and viral vectors for mAbs intratumor delivery systems. The figure shows the main advantages and limitations that distinguish viral and non-viral strategies.

**Table 1 ijms-24-02676-t001:** Summary of the most commonly used nanobodies for solid tumors, ongoing in clinical trials.

Drug Name (s)	Format	Target (s)	Type of Cancer	Current Status	ClinicalTrials.gov Identifier	Nº Participants	FDA Approval Status
KN046	Tetravalent, bispecific, Fc-fusion protein	CTLA-4, PD-L1	Advanced solid tumors and lymphoma	Phase II/III	NCT03872791	52	No
					NCT04474119	482	No
					NCT04925947	29	FDA-regulated Drug Product
Envolimab KN035	Monospecific, Fc-fusion protein	PD-L1	Advanced solid tumors, multiple primary neoplasm	Phase II	NCT03667170	200	No
					NCT04182789	20	No
					NCT04891198	200	No
αPD1-MSLN-CAR T cells	single-chain variable fragments (scFv)	αPD1-MSLN-CAR T	MSLN-positive Advanced Solid Tumors	Early phase I	NCT05373147	21	No
INBRX-109	Tetravalent, monospecific,	Death receptor 5	Advanced solid tumors, conventional chondrosarcoma	Phase I/II	NCT03715933	240	FDA-regulated Drug Product
	Fc-fusion protein				NCT04950075	201	FDA-regulated Drug Product
KN044	Monospecific, Fc-fusion protein	CTLA-4	Advanced solid tumors	Phase I	NCT04126590	39	FDA-regulated Drug Product

**Table 2 ijms-24-02676-t002:** Ongoing clinical trials of mAbs delivery.

Target (s)	Delivery	Type of Cancer	Current Status	ClinicalTrials.gov Identifier	Nº Participants	FDA Approval Status
PD-1	Intravenous infusion	Advanced gastric adenocarcinoma	Phase II	NCT03704246	123	No
	Intravenous injection	Gastric cancer	Phase I	NCT03713905	400	No
	Intravenous injection	Colorectal cancer.	Phase I/ II	NCT03711058	54	FDA-regulated Drug Product
	Intravenous infusion	Advanced Solid Tumors	Phase I/ II	NCT04775680	60	No
	Intravenous injection	Gastric cancer	Phase II	NCT03704246	123	No
	Intravenous injection	Advanced solid tumors	Phase I	NCT04478461	21	No
	Intravenous injection	Advanced refractory solid tumors.	Phase I	NCT02791334	215	FDA-regulated Drug Product
PD-1/TIM-3 Bispecific Antibody	Intravenous injection	Advanced and/or metastatic solid tumors	Phase I	NCT03708328	134	FDA-regulated Drug Product
Anti-CD47/PD-1 Bifunctional Antibody	Intravenous injection	Advanced solid tumors	Phase II	NCT04886271	210	No
PD-1/VEGF Bispecific Antibody	Intravenous infusion	Solid tumors	Phase I/ II	NCT04597541	59	No
PD-1/CTLA-4 Bispecific Antibody	Intravenous injection	Advanced or metastatic solid tumors	Phase I/ II	NCT03852251	338	No
PD-1 formulated with MK-5180	Subcutaneous Injection	Advanced Solid Tumors	Phase I	NCT05017012	72	No
CTLA-4	Intravenous injection	Advanced Solid Tumors	Phase I	NCT03849469	242	FDA-regulated Drug Product
CD39	Intravenous infusion	Locally advanced or metastatic solid tumors	Phase I	NCT05075564	60	FDA-regulated Drug Product
OX40	Intratumoral or intravenous injection	Advanced solid tumors	Phase I	NCT03831295	12	FDA-regulated Drug Product
LAG3	Intravenous injection	Advanced solid tumors	Phase I/ II	NCT01968109	1499	FDA-regulated Drug Product
CCR5	Subcutaneous Injection	Locally advanced or metastatic solid tumors	Phase II	NCT04504942	30	FDA-regulated Drug Product
4-1BB	Intravenous infusion	Advanced Solid Malignancies	Phase I	NCT04144842	50	No
PD-L1xCD27 Bispecific Antibody	Intravenous infusion	Advanced Solid Malignancies	Phase I	NCT04440943	27	FDA-regulated Drug Product
PD-L1	Intravenous infusion	Advanced Solid Malignancies	Phase I	NCT03590054	35	FDA-regulated Drug Product
Anti-PD-L1/Anti-CTLA4	Intravenous injection	Advanced Solid Malignancies	Phase I/II	NCT03518606	150	No

**Table 3 ijms-24-02676-t003:** Ongoing trials of intratumoral mAbs delivery.

	Target	Non i.t /co-i.t Therapy	Type of Cancer	ClinicalTrials.gov Identifier	Nº Participants	FDA Approval Status
	CTLA-4	Injection of ipilimumab during a biopsy procedure.	Head and neck Cancer	NCT02812524	18	FDA-regulated Drug Product
		Combination with intravenous nivolumab	Melanoma	NCT02857569	90	Not provided
		Combination with intravenous nivolumab	Glioblastoma	NCT03233152	6	No
		Intratumoral Tilsotolimod combination with intratumoral ipilimumab and intravenous nivolumab.	Advanced cancers	NCT04270864	72	No
	PD-1	mRNA-2752, a lipid nanoparticle encapsulating mRNAs encoding human OX40L, IL-23, and IL-36γ.	Ductal Carcinoma in Situ (DCIS)	NCT02872025	48	FDA-regulated Drug Product
		Intra-lesional nivolumab therapy	Cutaneous Kaposi Sarcoma	NCT03316274	12	FDA-regulated Drug Product
		Pre-operative cemiplimab administered intralesionally	Cutaneous Squamous Cell Carcinoma	NCT03889912	61	FDA-regulated Drug Product
		Combination of PD-1 and CTLA4	Metastatic Prostatic Adenocarcinoma	NCT04090775	12	FDA-regulated Drug Product
mAbs delivery and	CD40	Alone intratumorally or intravenously administered ADC-1013	Advanced Solid Tumors	NCT02379741	24	Not provided
non-viral theraphy		APX005M in Combination with systemic prembrolizumab	Metastatic Melanoma	NCT02706353	41	FDA-regulated Drug Product
		ABBV-927 and ABBV-181	Advanced solid tumors	NCT02988960; NCT03818542	163;3	FDA-regulated Drug Product
		Intratumoral Selicrelumab with atezolizumab	Relapsed B Cell Lymphoma	NCT03892525	4	No
		Fc-engineered anti-CD40 agonist	Lesions to the Skin	NCT04059588	28	FDA-regulated Drug Product
		SL-172154: fusion protein SIRPα-Fc-CD40L	Squamous Cell Carcinoma: Head and Neck or Skin	NCT04502888	5	FDA-regulated Drug Product
		D2C7-IT in Combination With 2141-V11	Malignant Glioma	NCT04547777	30	FDA-regulated Drug Product
		Intratumoral TriMix Injections (CD40 and CD27)	Breast Cancer Patients	NCT03788083	36	No
	OX40	mRNA 2416 alone or in combination with durvalumab	Advanced Malignancies	NCT03323398	79	FDA-regulated Drug Product
		Combinaiton with TLR9 agonist SD-101 and radiation	Low-Grade B-Cell Non-Hodgkin Lymphomas	NCT03410901	15	FDA-regulated Drug Product
	CD137	Urelumab combined with nivolumab	Solid Tumors	NCT03792724	32	Product Manufactured in and Exported from the U.S
	CD40	MEM-288:oncolytic adenovirus vector encoding transgenes for human IFNβ and a recombinant chimeric form of CD40-ligand	Solid tumors	NCT05076760	18	FDA-regulated Drug Product
	CD40	AdCD40L is a replication-deficient virus carrying the gene for CD40 ligand	Melanoma	NCT01455259	30	Not provided
Viral therapy	PD-1	MVR-C5252: oncolytic vectors expressing IL-12 and anti-PD-1 antibody	Recurrent or progressive glioblastoma	NCT05095441	51	FDA-regulated Drug Product
	PD-1	ONCOS-102: Oncolytic Adenovirus Expressing GMCSF and combined with prembrolizumab	Melanoma progressing after (PD1) Blockade	NCT03003676	21	Not provided
	CTLA-4	ISI-JX: Pexa-Vecan oncolytic virus genetically modified to express GM-CSF with intratumoural administration of ipilimumab	Advanced /solid tumors	NCT02977156	22	Not provided
	PD-1 and CTLA-4	ONCR-177 (which encodes CCL4, IL-12, Flt3L, anti-CTLA-4, anti-PD1alone or combined with pembrolizumab	Advanced/metastatic solid tumors	NCT04348916	132	FDA-regulated Drug Product

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
