# Peer review of "Leading Edge: Intratumor Delivery of Monoclonal Antibodies for the Treatment of Solid Tumors"

_ijms, 2023, doi:10.3390/ijms24032676_

Round 1
Reviewer 1 Report
This review article by Blanco et al "Leading edge: intra-tumor delivery of monoclonal antibodies for treatment of solid tumors" represents a nice and comprehensive overview on the various aspects of intra-tumoral antibody treatment with emphasis the various available delivery systems for direct delivery of therapeutic antibodies or of delivery of non-viral or viral vectors for their expression in vivo. The manuscript is clearly structured and written.
Although the all major aspects are covered, the sections on non-viral delivery platforms may be rounded up by inclusion of needle-free injection systems in addition to electroporation and some broader review of polymer nanoparticles, such as polycation-based vectors (PEI, Polylysine, dendrimer etc), and the use of inclusion of tumor-targeting ligands
Author Response
Thank you so much for taking the time to leave us feedback. We appreciate the time and effort that you dedicated to providing feedback on our manuscript and are grateful for the insightful comments on and valuable improvements to our paper. We have incorporated the suggestions made by the reviewers. Those changes are highlighted in yellow within the manuscript. Please see below, for a point-by-point response (in red) to the reviewers’ comments. All page numbers refer to the revised manuscript file with tracked changes.
Although the all major aspects are covered, the sections on non-viral delivery platforms may be rounded up by inclusion of needle-free injection systems in addition to electroporation and some broader review of polymer nanoparticles, such as polycation-based vectors (PEI, Polylysine, dendrimer etc), and the use of inclusion of tumor-targeting ligands.
1.In the section 5.1 of non-viral vectors, we added a paragraph about PEI nanoparticles (Page 8, lines:272-276). In this topic, there aren’t further studies on polymer nanoparticles applied to intratumor delivery of mAbs.
2.We haven’t found more studies focusing on intratumoral electroporation of mAbs.
3.We added a section 5.6 regarding tumor targeting ligands (Page 9, lines: 336-340). In this field, there are a few articles about intratumoral administration of mAbs.
Reviewer 2 Report
In present review, authors discusses on the potential use on intra tumor delivery of monoclonal antibodies for solid tumors treatment. I have several reservations, my comments are appended as below:
1. It is vague to refer just ‘cancer’ authors should stick to certain pathologies.
2. I observe that authors mention immune checkpoints in introduction but do not specify the cells expressing them and the function in brief. Author’s may refer PMID: 33076303 and add a para.
3. Nanobodies: indicate the FDA approval status, in which cancer types it is mostly used. This can be a table.
4. Authors should also enlist ongoing clinical trial with in vivo mAb delivery in separate table.
5. Lipid nanoparticles: greatest struggle with nanoparticles is specific delivery to target organs. Authors should specify the targeting approaches
6. mAb administration: authors should quote on target organs. Is there any approach to treat the solid intracranial tumors where BBB permeability is main issue?
7. irAEs: authors should explain on the types of immune cells involved. In addition, does systemic delivery leads the drug to lymph nodes where antigen presentation goes?
8. Intratumor mAb delivery: authors should add a table mentioning tumor type/ FDA approval status/ no of patients if already studied. Also explicitly specify the cancer type. I observe that authors note a table, it should be added with no of patients included.
9. There should be future directions section
Author Response
Thank you so much for taking the time to leave us feedback. We appreciate the time and effort that you dedicated to providing feedback on our manuscript and are grateful for the insightful comments on and valuable improvements to our paper. We have incorporated the suggestions made by the reviewers. Those changes are highlighted in yellow within the manuscript. Please see below, for a point-by-point response (in red) to the reviewers’ comments. All page numbers refer to the revised manuscript file with tracked changes.
1.It is vague to refer just ‘cancer’ authors should stick to certain pathologies.
1.We have sticked to certain solid tumors that we have specified in each section.
2.I observe that authors mention immune checkpoints in introduction but do not specify the cells expressing them and the function in brief. Author’s may refer PMID: 33076303 and add a para.
2.We have specified the cells that express each IC (Page 2, lines: 57-61).
3.Nanobodies: indicate the FDA approval status, in which cancer types it is mostly used. This can be a table.
4.Authors should also enlist ongoing clinical trial with in vivo mAb delivery in separate table.
3 and 4. It’s a great idea to add a table about Nanobodies for solid tumors ongoing in clinical trials (Page 3). In addition, we added a table ongoing clinical trial with in vivo mAb delivery (Page 4).
5.Lipid nanoparticles: greatest struggle with nanoparticles is specific delivery to target organs. Authors should specify the targeting approaches
5.We have highlighted in each section the type of administration for each treatment.
6.mAb administration: authors should quote on target organs. Is there any approach to treat the solid intracranial tumors where BBB permeability is main issue?
6.There is some approaches to treat the solid intracranial tumors where BBB permeability is main issue. In fact, he majority of chemotherapeutic agents do not cross the BBB and ones that do are removed by the efflux protein, p-glycoprotein. In several articles NPs have showed increase in permeability . Further studies could consider the evaluation of other CNS diseases, which are limited in therapeutic treatments by the BBB.
7.irAEs: authors should explain on the types of immune cells involved. In addition, does systemic delivery leads the drug to lymph nodes where antigen presentation goes?
7.We have explained the types of immune cells involved in irAEs (Page 11, lines: 429-431).
8.Intratumor mAb delivery: authors should add a table mentioning tumor type/ FDA approval status/ no of patients if already studied. Also explicitly specify the cancer type. I observe that authors note a table, it should be added with no of patients included.
8.We have completed the Table 3 with FDA approval status and nº of patients (Page 11).
9.There should be future directions section
9.We have added future directions section (Page 13, lines:545-569).
Round 2
Reviewer 2 Report
All my concerns are addressed.